# Mutual inhibition among postmitotic neurons regulates robustness of brain wiring in *Drosophila*

Marion Langen[1,2,3†], Marta Koch[1,2], Jiekun Yan[1,2], Natalie De Geest[1,2], Maria-Luise Erfurth[4,5], Barret D Pfeiffer[6], Dietmar Schmucker[4,5], Yves Moreau[7], Bassem A Hassan[1,2,3,6]*

[1]Center for the Biology of Disease, VIB, Leuven, Belgium; [2]Center for Human Genetics, University of Leuven School of Medicine, Leuven, Belgium; [3]Doctoral Program in Molecular and Cognitive Neuroscience, Doctoral School of Biomedical Sciences, University of Leuven, Leuven, Belgium; [4]Vesalius Research Center, VIB, Leuven, Belgium; [5]Department of Oncology, University of Leuven School of Medicine, Leuven, Belgium; [6]Janelia Farm Research Campus, Howard Hughes Medical Institute, Ashburn, United States; [7]Department of Electrical Engineering, University of Leuven, Leuven, Belgium

**Abstract** Brain connectivity maps display a delicate balance between individual variation and stereotypy, suggesting the existence of dedicated mechanisms that simultaneously permit and limit individual variation. We show that during the development of the *Drosophila* central nervous system, mutual inhibition among groups of neighboring postmitotic neurons during development regulates the robustness of axon target choice in a nondeterministic neuronal circuit. Specifically, neighboring postmitotic neurons communicate through Notch signaling during axonal targeting, to ensure balanced alternative axon target choices without a corresponding change in cell fate. Loss of Notch in postmitotic neurons modulates an axon's target choice. However, because neighboring axons respond by choosing the complementary target, the stereotyped connectivity pattern is preserved. In contrast, loss of Notch in clones of neighboring postmitotic neurons results in erroneous coinnervation by multiple axons. Our observations establish mutual inhibition of axonal target choice as a robustness mechanism for brain wiring and unveil a novel cell fate independent function for canonical Notch signaling.

*For correspondence: Bassem.
Hassan@cme.vib-kuleuven.be

†Present address: Department
of Physiology, University of Texas
Southwestern Medical Center,
Dallas, United States

Competing interests: The
authors declare that no
competing interests exist

Reviewing editor: Andrea Brand,
University of Cambridge,
United Kingdom

## Introduction

Conceptually, coupling axonal target choice to neuronal cell fate determination has the advantage of generating highly reproducible wiring patterns. However, it has the evolutionary disadvantage of limiting individual variability, and thus efficient adaptation to environmental change. Furthermore, a strictly deterministic coupling has the significant developmental disadvantage of limiting cellular plasticity in response to inevitable genetic and epigenetic variability within the developing brain (*Muotri et al., 2005*, *2010*). The latter point is particularly critical in neuronal lineages that are intrinsically variable in cell number.

Notch signaling is a highly conserved crucial regulator of early nervous system development. The most well-established function for Notch signaling in neural development is the selection of neural progenitor cells through a process termed mutual or lateral inhibition (*Artavanis-Tsakonas et al., 1999*). In lateral inhibition, a group of equivalent neuroepithelial cells that initially express roughly the same Notch activity are eventually segregated in two groups defined by expression and activity levels

**eLife digest** The brains of all members of a species are similar, but not identical, and these differences are partly responsible for the range of behaviors displayed by individuals. The development of the nervous system is known to depend on the Notch signaling pathway, but the mechanisms that regulate the balance between fixed patterns of neuronal connectivity vs individual variability are largely unknown.

Notch proteins are transmembrane proteins, which means that they have one part inside the cell membrane and another outside the cell. When a ligand protein—such as a Delta ligand—binds to the part that is outside the cell, the Notch protein breaks in two and the part inside the cell travels to the nucleus, where it can influence the expression of genes.

Cells are selected to become neurons through a process known as mutual, or lateral, inhibition. When a Delta ligand belonging to one cell binds to the Notch receptor on a neighboring cell, the production of Delta ligands in the second cell is reduced. This amplifies any initial differences in the amount of Delta produced by each cell, and leads ultimately to them becoming distinct cell types.

Now, Langen et al. show that this same mechanism is reactivated at a later stage of development during wiring up of the visual system. They used the fruit fly (*Drosophila*)—a model organism with a fully sequenced genome and short intergeneration time—to study a group of brain cells known as dorsal cluster neurons. At the end of the fruit fly larval stage, these neurons extend long axons across the brain to the opposite hemisphere: however, it is unclear how the neurons decide which cells to form connections with.

Using genetically modified flies, Langen et al. showed that inhibiting Notch in a single dorsal cluster neuron caused that neuron to target a different cell: however, other neurons adjusted their choices accordingly so that the overall pattern of connections remained unchanged. Inhibiting Notch in a cluster of dorsal cluster neurons, on the other hand, disrupted the entire network, suggesting that Notch-mediated communication between neurons (via mutual inhibition) is needed to establish a robust wiring map.

Langen et al. suggest that evolution has favored a mechanism that ensures that the overall pattern of connections within a circuit is preserved, while individual connections differ from one species member to the next.

---

of the receptor Notch and its membrane-bound ligand Delta. Due to cell–cell communication among overlapping clusters of adjacent cells, small differences in Notch expression level are amplified. As a consequence, the majority of cells within these clusters express high Notch activity levels, whereas a minority of cells express low Notch activity. These cells then go on to differentiate as neuronal progenitors. A second important function for Notch signaling is the establishment of alternate binary cell fates of sibling neurons. In contrast to lateral inhibition, this function requires the asymmetric inheritance of the Notch antagonist Numb by one of the two sibling neurons in a terminal cell division (*Guo et al., 1996*; *Spana and Doe, 1996*; *Knoblich, 1997*). It has been proposed that early binary cell fate specification via Notch-Delta signaling results in alternative axon targeting. In the fly visual system, the R7 and R8 photoreceptors, which independently require Notch activity to be specified, make alternative axonal target choices that appear to be consequences of their different fates. Indeed, their canonical fate marker transcription factors Prospero and Senseless, respectively, are required for this differential targeting (*Morey et al., 2008*). In the olfactory system of *Drosophila*, sibling neurons target their axon to different central nervous system (CNS) glomeruli due to differential cell fate specification (*Endo et al., 2007*). A recent elegant extension of this work shows that chromatin modification of Notch targets by the Hamlet protein further diversifies cell fate and axonal targeting in olfactory neurons (*Endo et al., 2012*). Interestingly, in this work, the effects were not restricted to sibling neurons, but extended to neighboring non-sibling neurons, and the corresponding effects on axonal targeting were not strictly binary nor fully penetrant leading to speculation that Notch activity might have probabilistic effects on axonal targeting downstream of cell fate specification (*Schmucker and Hassan, 2012*). Finally, an extensive analysis of CNS neurons in the fly olfactory circuit shows that there are more diverse wiring morphologies than can be accounted for by the number of cell types (*Chou et al., 2010*), suggesting that post-fate mechanisms may be crucial in wiring the *Drosophila* CNS. What these post-fate

mechanisms are, and how they buffer intrinsic variability to generate stereotyped wiring patterns, is unknown.

Here we show that the dorsal cluster neurons (DCNs), a group of bilateral commissural higher order neurons in the visual system of *Drosophila* (*Hassan et al., 2000*) are intrinsically variable in number within and between individuals, yet generate a highly stereotyped wiring diagram. We combine high resolution quantitative anatomical analyses, computational modeling, and genetic dissection to show that Notch-mediated mutual inhibition acts late in development on postmitotic neurons to bias axonal target choice. Finally, we show that this mechanism ensures the reproducibility of the DCN connectivity pattern.

## Results

### DCN medulla axons are derived from subclusters of neighboring DCNs

DCNs are produced from a single embryonic neuroblast, and all express the same cell fate markers, namely the Atonal (*Hassan et al., 2000*) and Acj6 (see below) transcription factors. All postmitotic DCNs are efficiently labeled with the *ato-Gal4* driver starting at the third instar larval (L3) stage (*Figure 1A*). DCNs innervate two regions of the optic lobes, the lobula and the medulla. This binary axon target choice is a result of three distinct steps (*Srahna et al., 2006*). First, in late third instar larvae DCN axons grow en masse toward the developing medulla neuropil attracted by a Wnt5 signal. Second, at 25–50 hr after puparium formation (APF), most DCN axons begin to retract back to the lobula neuropil, likely in response to a repulsive fibroblast growth factor (FGF) signal (*Figure 1A′–A″*). Wnt and FGF have opposing effects on Jun N-terminal Kinase (JNK) activity, which is both necessary and sufficient for DCN axon growth. Third, a small subset of regularly spaced axons resist the repulsive signal, by an unknown mechanism, and remain in the medulla (*Figure 1A‴*). These observations hint that the pattern of DCN axons may arise by a sorting mechanism that generates two mutually exclusive DCN axon target choices: lobula vs medulla (*Figure 1B″*).

To gain insight into how the two choices arise during DCN development, we carried out careful quantification of the number of DCN neurons and their axons in a cohort of genetically identical flies. Briefly, DCNs were labeled with GFP, driven by *ato-Gal4*, and imaged. Each DCN soma and axon were assigned 3D coordinates and analyzed with the 'clusterdata' function in MATLAB (see the 'Materials and methods' section for imaging and DCN profile details). These analyses reveal a number of interesting features. First, all DCNs express the same cell fate markers, Atonal and Acj6 (*Figure 2A–B′*), as detected by antibody staining, already at L3 prior to initiating axon outgrowth towards the optic lobes. Second, although there are on average 38 DCNs producing a highly stereotyped pattern of 12 spaced axons in the medulla (*Figure 2C*), the DCN lineage itself is not fixed. The number of neurons varies (35–43 neurons) not only between different individuals but also between the right and left sides of the same individual (r = −0.148; *Figure 2D*). Furthermore, there is no correlation between the number of neurons in a cluster and the number of medulla axons it projects (*Figure 2—figure supplement 1*, top graph). Thus, the DCNs represent an intrinsically variable lineage of similar neurons.

Next, we investigated whether positional information may play a role in determining medulla vs lobula target choice. DCNs are born in a ventral to dorsal order (*Zheng et al., 2006*), and each DCN establishes one ipsilateral dendritic arbor and one contralateral axonal arbor (*Srahna et al., 2006; Nicolai et al., 2010*). Using the flip-out technique (*Wong et al., 2002*), we labeled single DCNs with membrane-bound GFP and visualized them in the background of the entire cluster labeled with LacZ. We found that medulla targeting neurons are located all along the dorsal–ventral (D-V) and proximal–distal axes of the cluster (*Figure 2E–G*). Thus, there is no correlation between birth order and medulla vs lobula targeting, confirming earlier observations that sibling DCN clones generated using the MARCM technique can target the medulla, the lobula, or one each (*Zheng et al., 2006*). Interestingly, however, the D-V position of an axon is directly and very highly correlated (r = 0.964) with the D-V position of its soma within the cluster (*Figure 2H* and *Figure 2—figure supplement 1*, bottom graph) such that the most dorsal neuron gives rise to the most dorsal axon and so on (*Figure 2H*). Because most DCNs at any given D-V position innervate the lobula and a few innervate the medulla, D-V positional information does not directly instruct lobula vs medulla target choice.

We wondered whether cell–cell interactions might offer insight into the mechanism underlying the generation of a stereotyped medulla innervation pattern, despite variability in neuronal number. To this end, DCN soma were labeled with LacZ and their axons with CD8-GFP. Next, each neuron and each medulla axon were assigned their three-dimensional (3D) Cartesian coordinates (*Figure 3A–D*

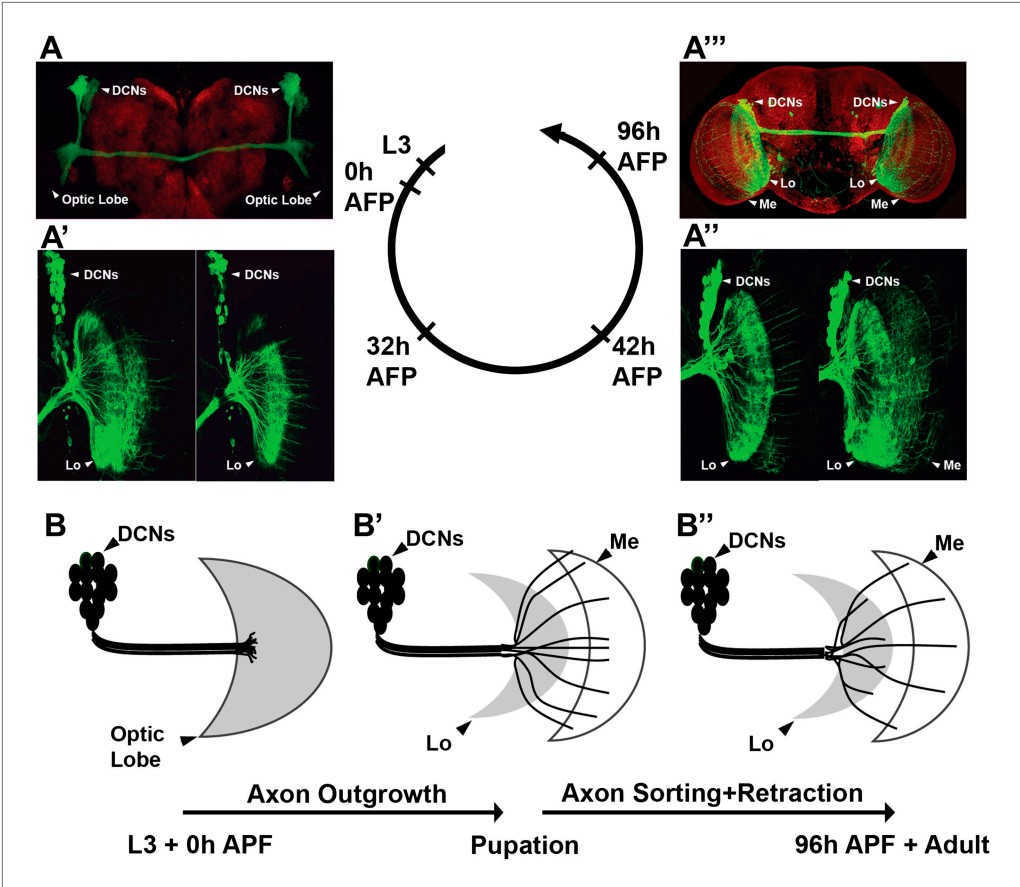

**Figure 1**. Development of the DCN wiring pattern. The development of the DCN cluster during various time points. At L3 (**A, B**) the neurites of the DCNs innervate the optic lobe. (**A'**, **A"**, **B'**) During early pupation, DCN axons innervate the lobula and extend towards the medulla (32 hr APF). (**A"**) After a retraction process, axons which remain in the medulla start to branch while the other axons retract back to innervate the lobula (42 hr APF). (**A'''**, **B"**) At adult stage the axonal pattern of the DCNs is complete (96 hr APF). APF: after puparium formation; DCN: dorsal cluster neurons.

and *Figure 3—figure supplement 1*) to determine the distances between DCN cell bodies, and between each two adjacent medulla axons (see 'Materials and methods'). These analyses reveal two features. First, DCN soma show a few sparse cells at the dorsal and ventral extremities and a majority of cells densely packed medially (*Figure 3C,D*). Second, DCN medulla axons are more dense dorsally and ventrally, but more sparse medially (*Figure 3B*). Therefore, there is an inverse correlation between cell density and interaxonal distance along the D-V axis in the medulla: four to eight medial DCN soma provide one medulla axon, whereas only one to two DCN soma target one axon to the medulla dorsally and ventrally (*Figure 3E*). It is crucial to note that this inverse relationship between soma density and medulla axon density is conserved in response to changes in cluster shape (*Figure 3—figure supplement 2*), yet the overall wiring pattern is preserved, strongly suggesting that DCNs use a dedicated mechanism to ensure the accuracy of the stereotyped medulla innervation pattern.

Together, the combination of data thus far suggests that the probability of any neuron targeting the medulla is independent of its absolute position in the cluster, but negatively dependent on the number of its neighboring neurons. One hypothesis to explain these observations stipulates the following: DCNs operate as several small overlapping subclusters in which all neurons within a certain distance of each other compete to provide one medulla axon (*Figure 3F*). In this model, neurons with fewer neighbors, as often seen dorsally and especially ventrally, target the medulla with higher probability. Conversely, densely clustered neurons, as seen medially, target the medulla with lower probability, and thus fewer axons. What genetic mechanism could underlie such a selection process?

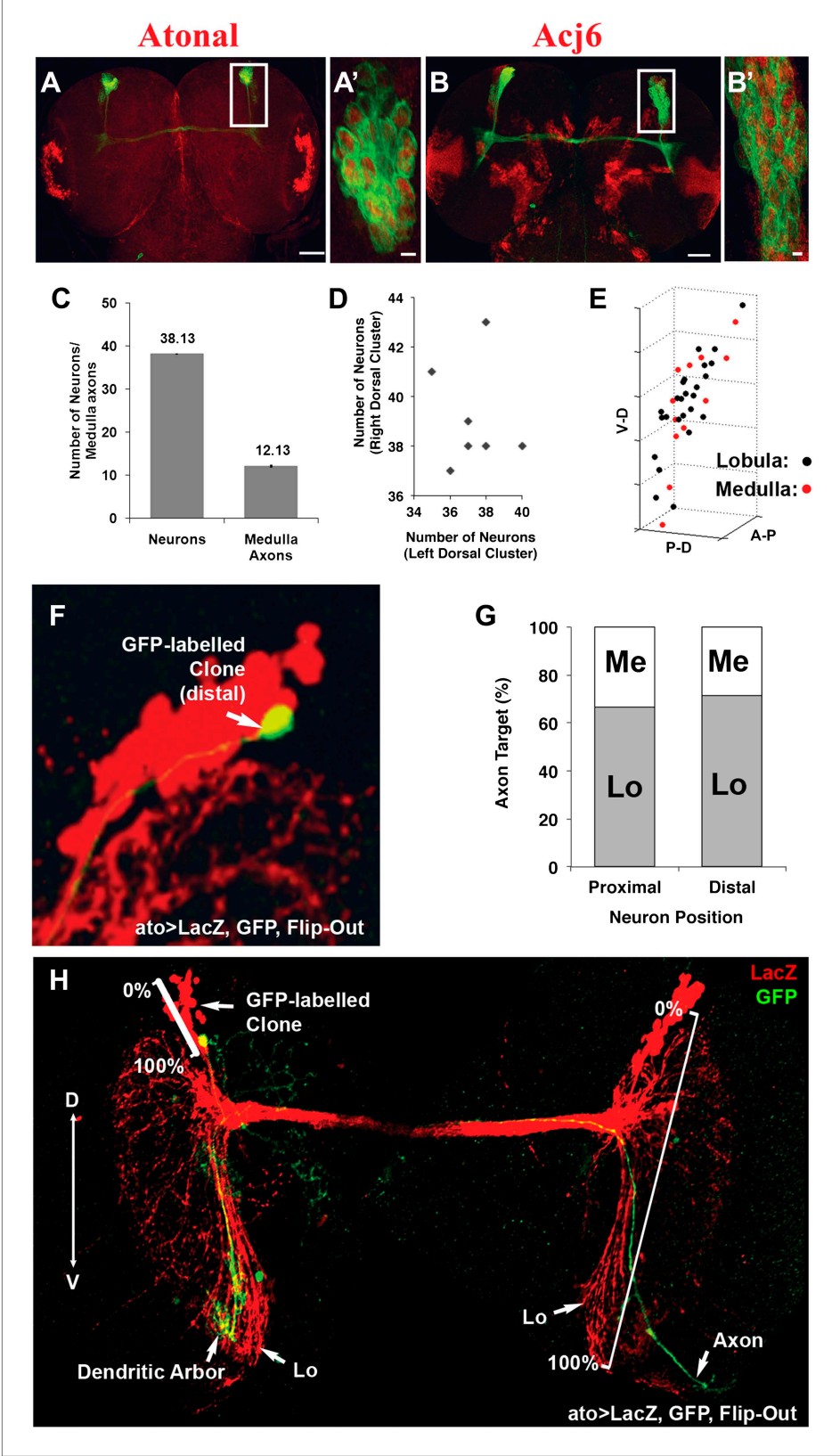

**Figure 2**. DCNs are equivalent, their lineage is intrinsically variable and does not predict the wiring pattern. (**A, A'**) The cell fate marker atonal is expressed at L3 stage in all DCNs. (**B, B'**) Acj6, a cell fate marker is expressed in all DCNs at L3 stage. (**A, B**) Scale bar: 50 μm. (**A', B'**) Scale bar: 5 μm. (**C**) Quantification of the number of DCN soma
*Figure 2. Continued on next page*

*Figure 2. Continued*

(38.13 ± 2 SD, n = 15) and medulla axons (12.13 ± 1.36 SD) (Data shown as mean ± SEM). (**D**) There is no correlation detectable between the number of neurons of the left and right clusters in individual brains (r = −0.112, n = 14). (**E**) 3D reconstruction of a single dorsal cluster. Red cells symbolize neurons innervating the medulla. These neurons are distributed along the D-V axis. (**F**) An example of a single GFP-positive DCN soma located distally within the dorsal cluster which is marked in red (LacZ). (**G**) Quantification of the position of a DCN soma (proximal, distal) within the dorsal cluster and its axon target choice of medulla or lobula neuropil (n = 16). There is no preference for axon target choice based on the cell body position within the cluster. (**H**) An example of the flip out technique used to analyze the relation between neuron position and axon position along the D-V axis. DCN: dorsal cluster neurons; D-V: dorsal–ventral.

The following figure supplements are available for figure 2:

**Figure supplement 1**. DCN cell body position highly correlates with its D-V axon position.

## Notch signaling among DCN neurons is required for target choice

Conceptually, the model above is similar to the generation of neural precursor cells from small subsets of interacting ectodermal cells, also known as proneural clusters. Notch-mediated mutual inhibition gives rise to a few regularly spaced single neural progenitors while the majority of cells adopt the alternative epidermal fate (*Campos-Ortega and Jan, 1991*; *Artavanis-Tsakonas et al., 1999*). Indeed, DCNs express Notch and its ligand Delta during the time when DCN axon sorting is occurring (~32 hr APF; *Figure 4A–A'''*). At this stage, all DCNs also express the Delta regulator Neuralized (Neur) and the canonical Notch pathway transcriptional reporter E(spl)-CD2 (*Bailey and Posakony, 1995*; *Lecourtois and Schweisguth, 1995*; *Lai et al., 2001*; *Pitsouli and Delidakis, 2005*; *Figure 4B–C*). Strikingly, using anti-CD2 antibodies, we find differences in the levels of the Notch activity reporter among individual DCNs: most cells express relatively high levels of E(spl)-CD2, whereas a minority of cells scattered across the D-V axis of the cluster express relatively low levels (*Figure 4D–F'*). In contrast, E(spl)-CD2 levels are very low in DCNs at L3 (data not shown) suggesting that Notch signaling is not active in postmitotic DCNs prior to pupariation. Finally, we do not detect expression of the asymmetric inherited Notch inhibitor Numb in the DCNs nor a phenotype with *numb* RNAi expression (data not shown), suggesting that differences in Notch activity between DCNs are unlikely to arise by differential inheritance, a cell fate determinant. Together these data show that Notch signaling is active within the DCNs at the time of axon target sorting.

To examine the potential consequence of N-mediated mutual inhibition on an irregular 3D cluster of neurons, we first extended an existing computational model developed for mutual inhibition in a regular 2D hexagonal lattice (*Collier et al., 1996*) to quantitatively describe the emergence of DCN axon target choice from DCN cell clusters (see 'Materials and methods' and *Figure 5—figure supplements 1 and 2*). The model predicts medulla targeting by a similar number and pattern of DCN axons as observed in vivo (*Figure 5A* red cells, *Figure 5C*). Next, as no previous modeling of Notch has been performed under loss of function conditions, we tested reduced Notch activity in silico. In this case, the model predicts that the stereotyped number and pattern of DCN medulla axons are disrupted. Specifically, the medulla is targeted by an increased number of axons (17.29 ± 3.96) arising from several small clusters (2–3) of neighboring neurons, especially medially (*Figure 5B*, red cells and blue arrows, *Figure 5C*).

To test these predictions, we asked whether Notch signaling is required within the DCNs. We used the ato-Gal4 driver to inhibit Notch and Delta specifically in postmitotic DCNs during adult brain development. It is critical to note that during adult brain development, ato-Gal4 is only active in DCNs postmitotically and after their axons begin to grow (*Hassan et al., 2000*; *Srahna et al., 2006*; *Zheng et al., 2006*). Postmitotic, DCN-specific inhibition of Notch or its ligands Delta and Serrate (Ser) leads to two phenotypes. First, we observe clustered axons at several D-V positions, particularly medially, as opposed to isolated single axons seen under control conditions (*Figure 5D–F',I*) causing a significant decrease in axon–axon distances in the medulla (*Figure 5K*). Second, we observe a significant increase in the number of axons targeting the medulla (*Figure 5L* and *Figure 5—figure supplement 3*), indicating that axon clustering is the result of neighboring axons erroneously making similar target choices. These data quantitatively mirror the phenotypes predicted by computational modeling of mutual inhibition among DCNs. Importantly, loss of Notch function alters neither the

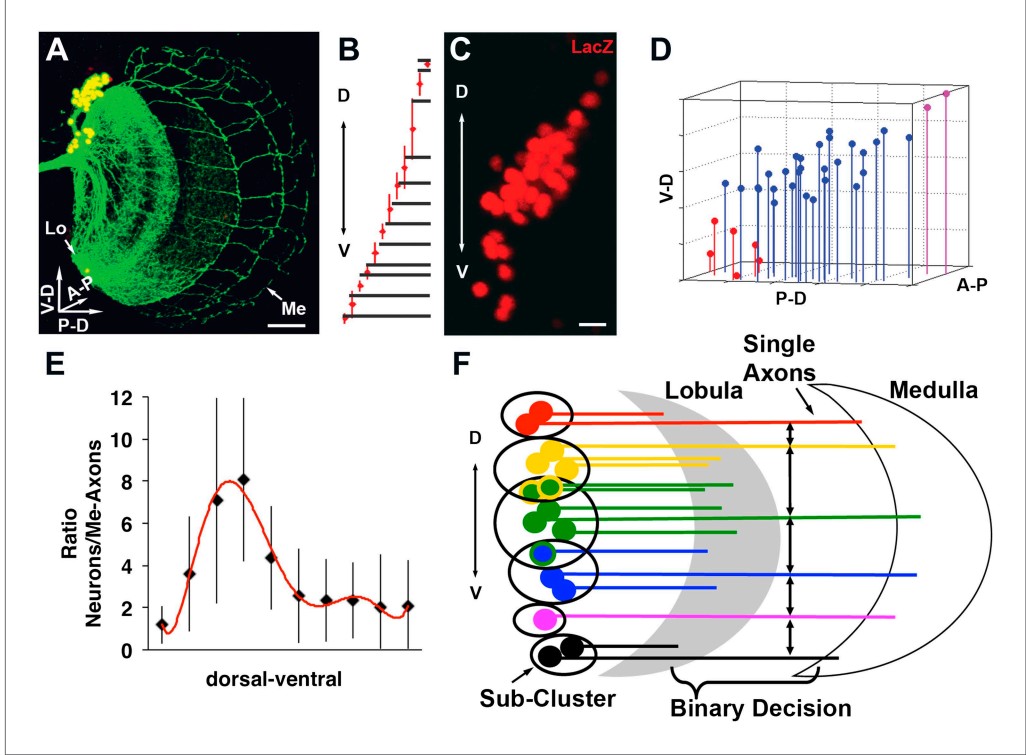

**Figure 3**. Quantitative analysis of the DCN soma and axon. (**A**) A typical DCN axonal pattern showing GFP-labeled neurites (green) and LacZ labeled soma (red). Scale bar: 20 μm. (**B**) Analysis of interaxonal distances of DCN medulla axons along the D-V and A-P axes. Medulla axons are less dense medially and more dense dorsally and ventrally (n = 16). (**C**) Projection of a typical DCN cluster with sparse soma dorsally and ventrally and a high soma density medially. Scale bar: 5 μm. (**D**) 3D stem plot of the DCN cluster shown in (**C**) displays lines extending form P-D and A-P axis while the 3D position is marked with a filled circle—red indicates ventral cells, blue shows medial, and magenta shows dorsal cells. Vertical lines along the P-D and A-P axes form a representation of the density of the DCNs. (**E**) Ratio of the number of DCNs and their medulla axons along the D-V axis (n = 16). While dorsally and ventrally 1–2 neurons generate one medulla axon, medially 4–8 neurons produce one medulla axon. (**F**) Schematic representation of the sub-cluster model: a DCN cluster is divided into several overlapping subsets of neighboring neurons. Each subset provides one medulla axon which leads to pattern of spaced single axons in the medulla. A-P: anterior–posterior; DCN: dorsal cluster neurons; D-V: dorsal–ventral; P-D: proximal–distal.

The following figure supplements are available for figure 3:

**Figure supplement 1**. Counting DCN soma and medulla axons.

**Figure supplement 2**. Variability of the DCN cluster shape induces robust wiring pattern.

number (38.13 ± 2 SD vs 37.12 ± 3.1 SD, *Figure 5—figure supplement 4*) nor the expression of DCN fate markers (*Figure 5—figure supplement 5*). RNAi knockdown of Neur, a positive regulator of Delta and Ser activity, causes clustered axons and a significant increase of medulla axons (*Figure 5G,L*). Finally, postmitotic and DCN-specific RNAi knockdown of the canonical Notch pathway nuclear effector proteins Suppressor of Hairless (Su(H)) and Mastermind (Mam) also result in axon clustering and a corresponding increase in medulla axon number (*Figure 5H–J',M*). Altogether, these data support the notion that DCNs require Notch-mediated mutual inhibitory interactions to make accurate medulla vs lobula axon target choices.

## Notch antagonizes JNK activity via Pak inhibition

Previous work has shown that high JNK activity is required for DCN axons to resist retraction and innervate the medulla. We hypothesized that N and JNK might interact negatively in neurons as described in *Drosophila* embryos and mammalian cell culture (*Zecchini et al., 1999*; *Chu et al., 2002*;

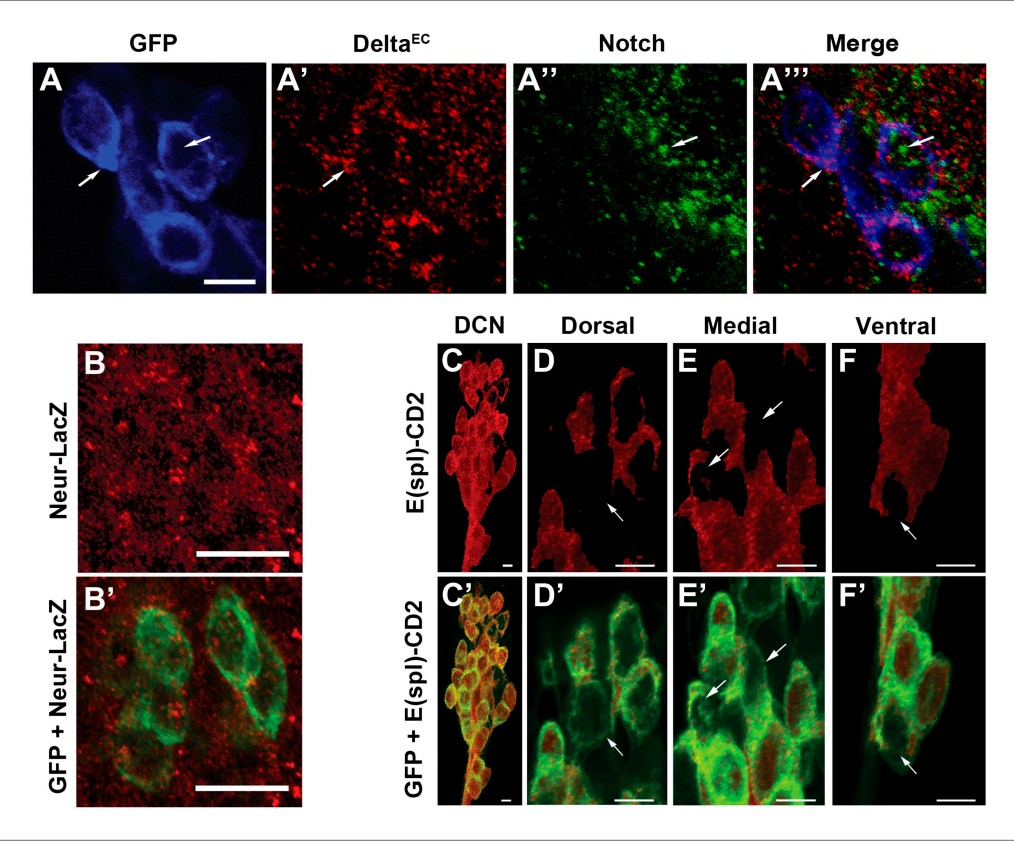

**Figure 4**. Canonical Notch pathway proteins are expressed during axon targeting. (**A–A'''**) Expression of Delta (red) and Notch (green) in GFP-labeled DCNs (blue) at ~32 hr APF. DCNs express both Delta and Notch and the proteins appear intracellular, indicative of active signaling. Scale bar: 5 µm. (**B, B'**) Expression of Neur-LacZ (red) in GFP-labeled DCNs (green) at ~32 hr APF. (**C–F**) Expression of E(spl)-CD2 reporter (red) in the DCN soma at ~32 hr APF. Within the dorsal cluster, there are DCN soma with different E(spl) expression levels (marked with white arrows). (**C'–F'**) Expression pattern of E(spl)-CD2 reporter (red) together with GFP. White arrows indicate DCN soma with little or no E(spl)-CD2 expression. Scale bar: 5 µm. DCN: dorsal cluster neurons.

*MacKenzie et al., 2004*; *Mateos et al., 2007*). Furthermore, it has been proposed that downregulation of the axon guidance molecule p21 activated kinease1 (Pak1) via canonical N signaling attenuates JNK activity (*Mateos et al., 2007*), making *Drosophila* Pak an attractive target candidate in our system. Indeed, we find that gain of Pak function, by expression of the constitutively active Pak using *ato-Gal4*, phenocopies Notch inhibition resulting in a significant increase and clustering of medulla axons, while knockdown of Pak levels reduces the number of medulla innervations (*Figure 6A–D*). Next, we examined the DCN medulla targeting pattern under conditions of inhibition of both N and JNK, which have opposite effects on medulla targeting. Simultaneous downregulation of JNK and N activities, using dominant negative transgenes driven by *ato-Gal4*, results in a phenotype similar to JNK inhibition alone (*Figure 6D*). This supports the notion that N activity antagonizes JNK signaling. Simultaneous reduction of both Notch activity and Pak levels, using Pak RNAi, results in a phenotype indistinguishable from Pak reduction alone (*Figure 6D*), supporting the notion that Pak acts downstream of Notch. Finally, DCN-specific gain of Pak and loss of JNK activity shows a reduced number of medulla axons, resembling loss of JNK (*Figure 6D*). These data support a model whereby N reduces JNK activity via inhibition of Pak function (*Figure 6E*). To test this model further, we asked if gain of Notch function affects JNK activity and Pak expression levels. To this end, we expressed the constitutively active intracellular domain of Notch (NICD) in DCNs and examined the levels of JNK reporter construct *puckered-LacZ* (*puc-LacZ*; *Reed et al., 2001*). We found a significant reduction of *puc-LacZ* levels in response to Notch activation (*Figure 6F*). Next, we transfected the *Drosophila* neuronal cell

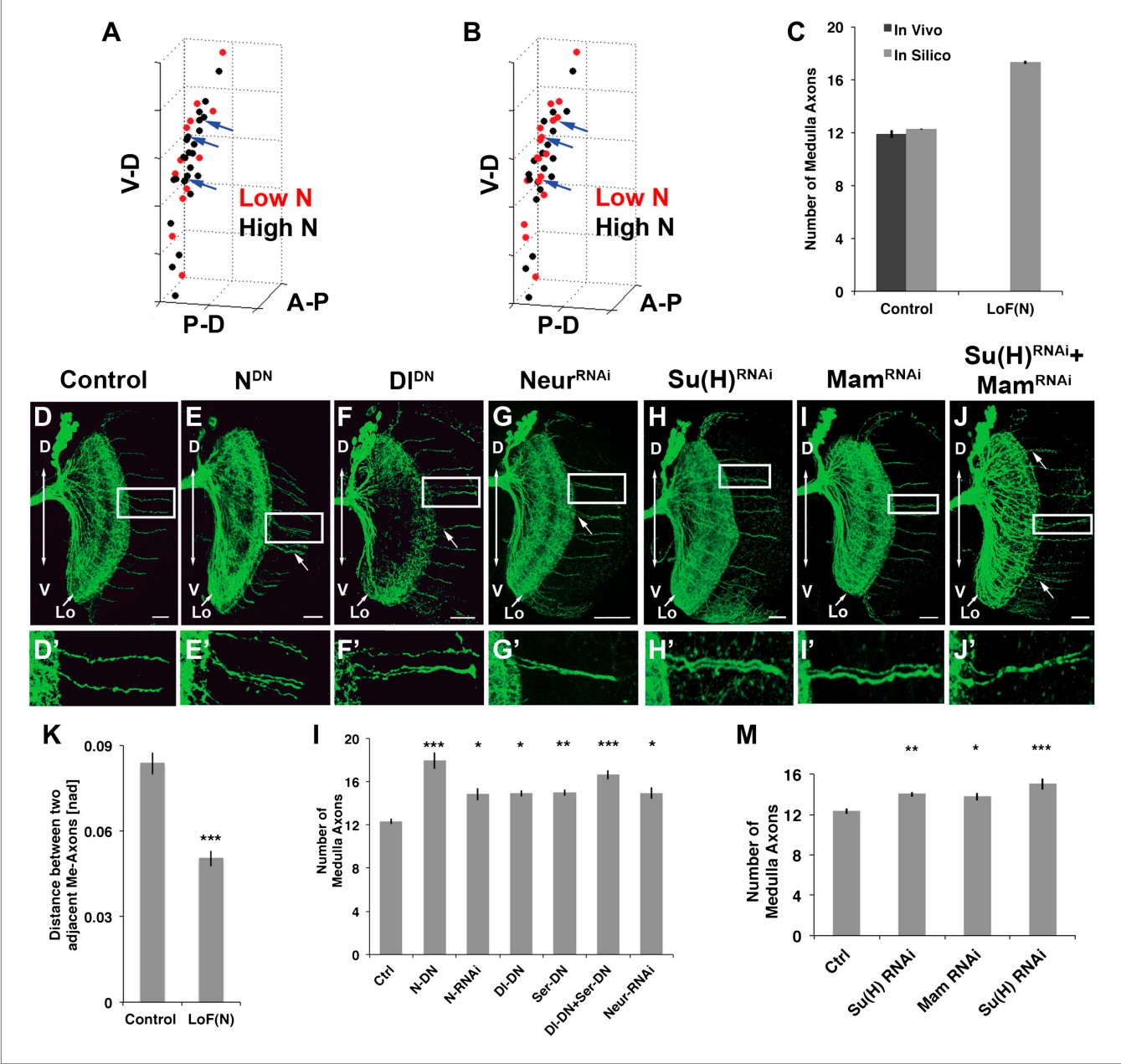

**Figure 5**. Canonical Notch pathway genes are required postmitotically for axon targeting. (**A, B**) A representation of an in vivo DCN cluster with in silico control (**A**) and in silico reduced Notch activity conditions (**B**). Red symbolizes cells with low Notch expression level. These cells will project their axon to the medulla. Black represents cells with high Notch expression level. Black cells will target their axon to the lobula. While in (**A**) red cells are singular (blue arrows) in (**B**) red cells located medially appear to be clustered (blue arrows). V-D: ventral–dorsal axis; P-D, proximal–distal axis; A-P, anterior–posterior axis. (**C**) The in silico predicted number of medulla axons under control resemble the in vivo data. In vivo control 11.88 ± 1.2 SD (n = 16), in silico control 12.26 ± 2.3 SD (p>0.05) (n = 1600). In silico loss of Notch function prediction results in an increase of medulla axons: 17.29 ± 3.96 SD (n = 1600). Data shown: mean ± SEM. (**D**) Control pattern of spatially separated single medulla axons. (**E-G, E'-G'**). Reduction of Notch or Delta activity or RNAi knock-down of Neur result in clustered axons. (**H-J, H'-J'**) RNAi knock-down of Su(H), Mam, or both results in clustered axons. Scale bar: 20 μm. (**K**) Quantification of the clustered axons observed under loss of Notch condition in comparison to control. Clustered axons were described by distance measurements between two adjacent medulla axons. Distances between two adjacent medulla axons under loss of Notch condition are significantly smaller. Control: 0.084 [nad] (=normalized arbitrary distance) ± 0.044 SD (n = 136), Notch[DN] 0.050 [nad] ± 0.034 SD (n = 174) (p<0.0001, Mann–Whitney Test, both distribution are non-Gaussian). (**L**) Inhibition of Notch, the ligands Delta and serrate, and Neuralized increases the number of medulla axons. Control 12.33 ± 0.89 SD (n = 12, Gaussian distribution), Notch[DN] 17.95 ± 2.67 SD (n = 20, Gaussian distribution) (p<0.001), Notch RNAi 14.83 ± 1.9 SD (n = 12, Gaussian distribution) (p<0.05), Delta[DN] 14.92 ± 1 SD (n = 12, Gaussian distribution) (p<0.05), Serrate[DN] 15.0 ± 1.62 SD (n = 16, Gaussian distribution) (p<0.01), Delta[DN] + Serrate[DN] 16.64 ± 1.92 SD (n = 14, Gaussian distribution) (p<0.001), Neur RNAi 14.93 ± 1.11 SD (n = 15, non-Gaussian distribution)

*Figure 5. Continued on next page*

*Figure 5. Continued*

(p<0.05); (non-parameteric Kruskal–Wallis Test, Data shown: mean ± SEM). (**M**) Inhibition of Su(H), Mam, or both significantly increases the number of medulla axons. Control 12.33 ± 0.89 SD (n = 12, Gaussian distribution), Su(H) RNAi 14.00 ± 0.82 SD (n = 10, Gaussian distribution) (p<0.01), Mam RNAi 13.75 ± 1.22 SD (n = 12, non-Gaussian distribution) (p<0.05), Su(H) RNAi + Mam RNAi 15.0 ± 1.66 SD (n = 9, non-Gaussian distribution) (p<0.001). Non-parameteric Kruskal–Wallis Test, Data shown: mean ± SEM. DCN: dorsal cluster neurons; SD: standard deviation.

The following figure supplements are available for figure 5:

**Figure supplement 1**. Relation between DCN soma distance and the number of adjacent cells within a subcluster

**Figure supplement 2**. Relation between distances of DCN soma and number of medulla axons.

**Figure supplement 3**. Loss of Delta increase the number of medulla axons.

**Figure supplement 4**. No function of Notch in DCN cell number.

**Figure supplement 5**. Cell fate marker is expressed in every single neuron at L3.

line DmBG2-c3 with NICD and assayed the effects on Pak and phospho-JNK (active JNK) levels. This resulted in a significant decrease in both Pak and pJNK levels, consistent with our in vivo observations (*Figure 6G*). Together, these data suggest that Notch activity sets the baseline JNK activity threshold which distinguishes medulla-targeting from lobula-targeting neurons (*Figure 6E*).

## Notch signaling is required during the axonal sorting phase

The data above suggest that Notch signaling is required to segregate axon target choices sometime between L3 and adult. To determine more precisely the timing of Notch requirement, we took advantage of the dominant nature of loss of Notch function and a temperature sensitive Notch mutant allele (Notch^ts). Flies were raised at either the permissive (18°C) or restrictive temperature (28°C) between 0 and 96 hr APF (*Figure 7A,B*). Reduction of Notch activity at the restrictive temperature results in clusters of neighboring axons in the medulla and a significant increase in the number of medulla axons (*Figure 7A–E*). Next, we raised flies at the permissive temperature and shifted them to the restrictive temperature for 24 hr intervals at different APF stages (*Figure 7A,B*). Decreasing Notch activity only alters axon target choice between 24 and 60 hr APF, but neither before nor after (*Figure 7B,E*). Importantly, inhibition of Notch function specifically within the DCNs only during pupal development, using a combination of ato-Gal4 and a temperature-sensitive Gal80 (*McGuire et al., 2003*), results in a very similar phenotype (*Figure 7E'*). Finally, gain of Notch function via DCN-specific expression of NICD or Su(H) strongly reduces the number of medulla axons found in the adult brain, but, importantly, does not inhibit their initial outgrowth into the medulla (*Figure 7F–J*), confirming that canonical Notch signaling specifically regulates the ability of axons to remain in the medulla after initial targeting. Altogether, these data mean that Notch signaling is required within the DCNs specifically to segregate axon target choice at pupal stage between 24 and 60 hr APF, but neither before nor after.

## Notch acts cell autonomously to bias axonal target choice

A mutual inhibition model for axonal target choices makes three specific predictions (*Figure 8A*). First, loss of Notch in a single neuron should autonomously bias the target choice of the mutant neuron. Second, the process should be robust such that a change of target choice by the Notch mutant neuron should be compensated for by a neighboring neuron. Thus, the total number and spacing of medulla axons should remain within wild-type range. In contrast, loss of Notch activity in subclusters of neighboring neurons result in multiple axons from the same subcluster innervating the medulla. We set out to systematically test these predictions.

First, we asked if Notch acts cell autonomously by generating GFP-labeled single cell clones mutant for Notch function using the MARCM technique (*Lee and Luo, 1999*). We generated two types of clones: clones in which Notch activity is inhibited only in postmitotic DCNs specifically after L3 using dominant negative Notch, and Notch-null clones in which Notch is completely absent from single cells from the time they are born. We found that whereas only 24% of the control clones target the medulla,

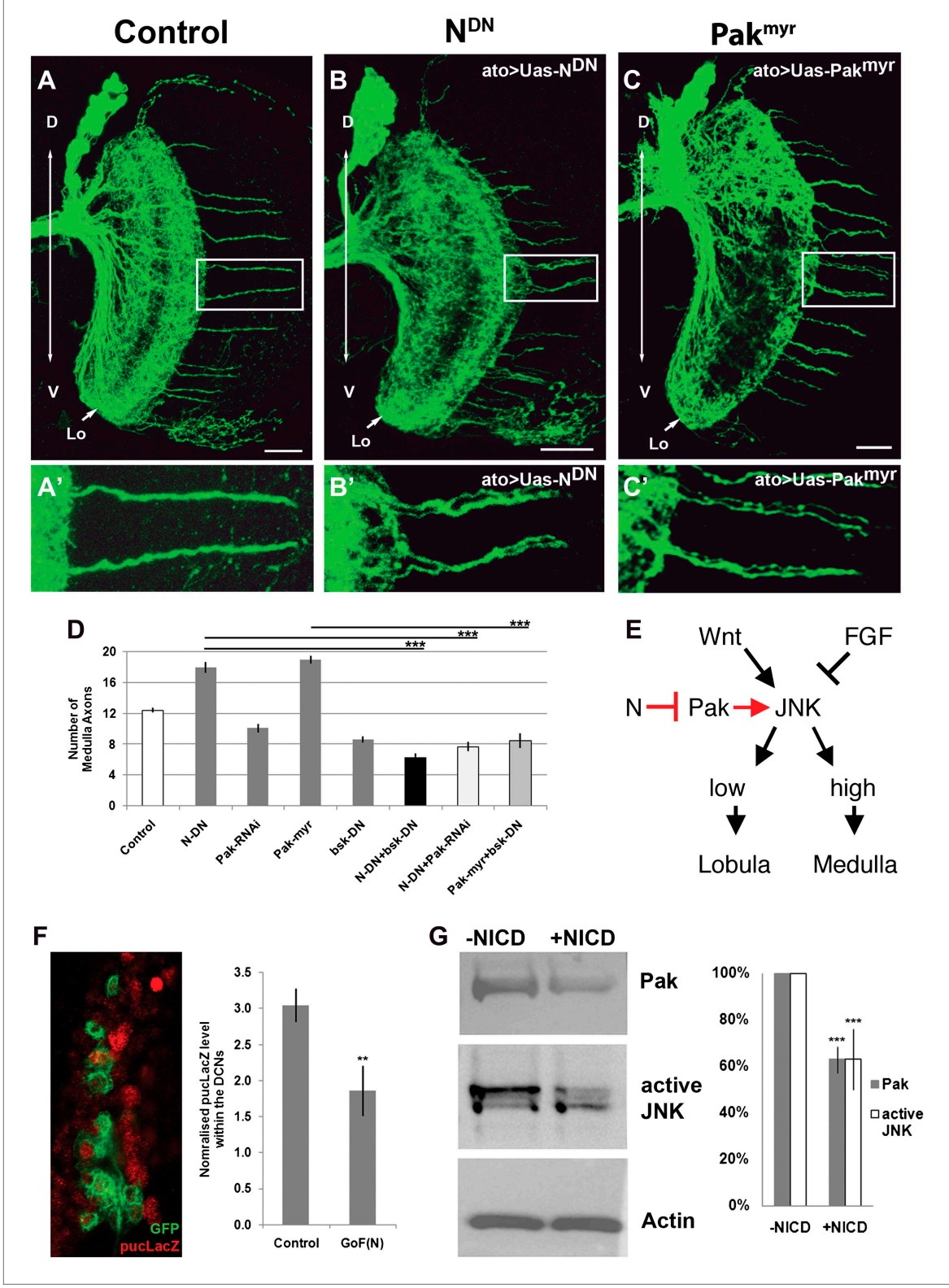

**Figure 6**. Notch signaling attenuates JNK activity via Pak (**A–C'**) loss of Notch signaling as well as constitutive Pak activation result in clustered axons. Data shown: mean ± SEM. Scale bar: 20 μm. (**D**) Analysis of the number of medulla innervations under the following conditions: Control 12.33 ± 0.89 SD, Notch[DN] 17.95 ± 3.19 SD (p<0.05 to Control), Pak RNAi 10.1 ± 1.73 SD, Pak[myr] 19 ± 1.8 SD (p<0.05 to control), bsk[DN] 8.6 ± 1.26 SD, Notch[DN] and bsk[DN] 6.3 ± 1.57 SD (p<0.001 to Notch[DN]), Notch[DN] + Pak RNAi 9.25 ± 2.38 SD (p<0.001 to Notch[DN]), and Pak[myr] + bsk[DN] 8.5 ± 2.35 SD (p<0.001 to Pak[myr]).
*Figure 6. Continued on next page*

*Figure 6. Continued*

(**E**) Schematic of the interaction between Notch, Pak and JNK signaling to determine axonal target choice. (**F**) A subset of DCNs marked with GFP (green) and pucLacZ (red) shown in the left panel. Analysis of pucLacZ in control and gain of Notch function DCNs (3.04 ± 0.24 SD vs 1.86 ± 0.35 SD, n = 3, (p=0.01, Student's t-test)). pucLacZ level within the DCNs were normalized by measurements of pucLacZ level of the entire image except the DCNs. (**G**) The effects of Notch gain of function (by transfection of NICD) on Pak expression and JNK activity in *Drosophila* neuronal cell line DmBG2-c3. Western blot and quantification of the effects on Pak (1.0 ± 0.0 vs 0.628 ± 0.05 SD [p<0.001] Student's t-test) and phospho-JNK (active JNK) levels (1.0 ± 0.0 vs0.63 ± 0.129 SD [p<0.001, Student's t-test]). DCN: dorsal cluster neurons; NICD: intracellular domain of Notch; SD: standard deviation; SEM: standard error of the mean.

approximately 60% of all Notch mutant clones—both null and DN—do so, with the most pronounced effect occurring medially (p<0.001 null, p<0.001 DN, Fisher's exact test) (*Figure 8B–D*). In contrast, as expected, medial Delta mutant clones favor targeting the lobula (*Figure 8B'*) with little or no effect on dorsal or ventral clones (data not shown). Therefore, Notch signaling levels autonomously regulate the axon target choice of individual DCNs. These data validate the first prediction of the mutual inhibition hypothesis.

To test the second prediction, we labeled the entire DCN pattern with the red fluorescent protein Cherry and generated single GFP-labeled Notch mutant cells in that background (*Figure 8E*). Analysis of the number, clustering, and spatial pattern of medulla axons in these clones show that the Notch mutant axon remains correctly spaced and the overall number and pattern of axons in the medulla is indistinguishable from control (*Figure 8E',F* and *Figure 8—figure supplement 1*). Next, we analyzed 23 axon–axon distances representing 12 individual mutant clones and compared them to the distances observed in controls. We found that 87% (20/23) of the mutant axons are within the normal range, as expected if neighboring control neurons innervate the lobula (*Figure 8G*). Crucially, the biased medulla innervation caused by loss of Notch activity does not alter the D-V positional relationship between neuronal soma and axon; dorsal neurons still project dorsally, medial neurons project medially, and ventral neurons project ventrally (*Figure 8B* and *Figure 8—figure supplement 2*).

## A mutual inhibition mechanism of axonal target choice ensures accuracy of neuronal connectivity

To test the third prediction, we designed a novel genetic combination to create adjacent cell clones independently of birth order. To this end, we generated flies in which Notch activity can be stochastically manipulated in sets of 'green' (GFP-positive) postmitotic DCNs, while their 'red' (Cherry-positive) neighbors remain wild type (see 'Materials and methods' for genotype details) in order to inhibit Notch function during axonal targeting in potentially interacting DCN subclusters. As controls, subsets of adjacent wild-type DCNs were labeled with GFP without inhibition of Notch activity. Of 20 control clones, 18 (90%) gave rise to normal medulla innervation patterns (*Figure 8H–H''*) with no effect on the number of medulla axons (*Figure 8J*). In two clones (10%), we observed one example of two adject axons each. In contrast, when Notch is stochastically inactivated in subsets of postmitotic DCNs, two types of phenotypes are observed. First, in 55% of all clones (24/44), the overall pattern remains normal, even though the mutant axons themselves target the medulla. Second, in 45% of the clones (20/44), in addition to mutant axons targeting the medulla (*Figure 8I'*, white arrows), axon clustering is observed (p=0.0094 Fisher's exact test). Furthermore, the number of medulla axons is significantly increased, and the axon–axon distances within these clones is drastically reduced (*Figure 8I–K* and *Figure 8—figure supplement 3*), suggesting that in these cases interacting neurons of one subcluster have been targeted by the stochastic inhibition of Notch. These data validate the third prediction and strongly support the idea of Notch-mediated interactions among neighboring cells.

## Discussion

Previously, two deterministic models have been proposed to explain the requirement for Notch signaling in brain connectivity. First, binary sibling neuron fate specification by Notch—as opposed to mutual inhibition—has been proposed as a mechanism of fate-dependent targeting of sensory axons (*Endo et al., 2007*). Notch has also been suggested to act as an axon guidance molecule via a noncanonical pathway in fly embryos (*Crowner et al., 2003*). We show that higher order CNS neurons utilize mutual inhibition via Notch signaling to establish a robust wiring map. However, in contrast to the

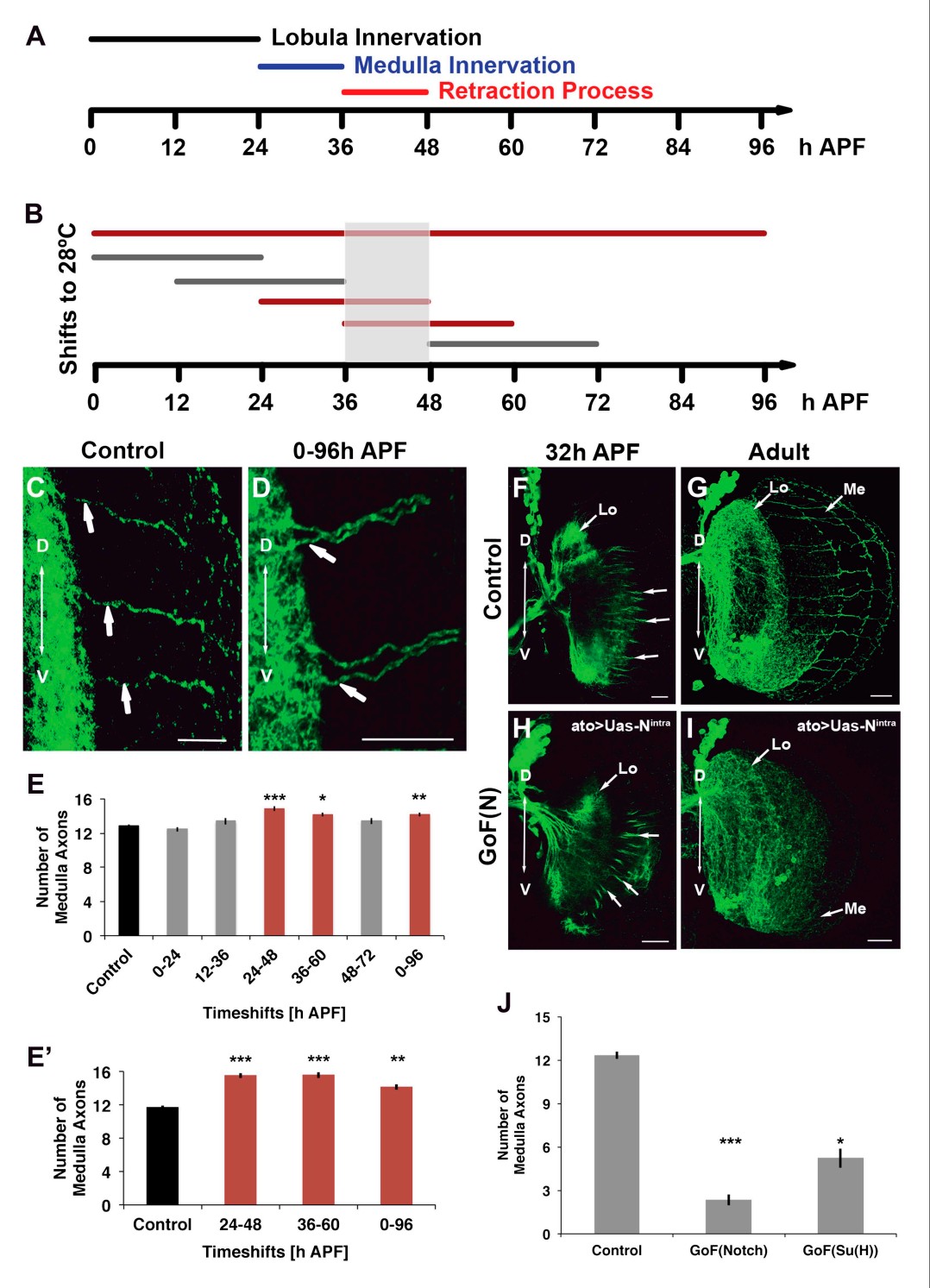

**Figure 7**. Notch acts specifically during the axon sorting phase to regulate DCN axon targeting. (**A**) Schematic of the axonal dynamics of DCNs during pupal development. (**B**) Schematic of the temperature shift experiments (to 28°C) to determine the temporal requirement for Notch during pupal development. (**C–D**) Reduction of Notch activity (using Notch$^{ts}$) during all of pupation leads to clustered axons in comparison to control. Scale bar: 20 μm. (**E**) Reduced Notch activity (N$^{ts}$) alters the medulla innervation pattern at 24–60 hr APF, but neither before nor after. Control 12.86 ± 0.66 SD (n = 14, non-Gaussian distribution), 0–96 hr APF 14.18 ± 0.92 SD (n = 18, non-Gaussian distribution) (p<0.01), 0–24 hr APF 12.46 ± 0.97 SD (n = 13, Gaussian distribution), 12–36 hr APF 13.38 ± 1.39 SD (n = 14, non-Gaussian distribution), 24–48 hr APF 14.86 ± 1.03 SD (n = 14, non-Gaussian distribution) (p<0.001),
*Figure 7. Continued on next page*

*Figure 7. Continued*

36–60 hr APF 14.15 ± 0.69 SD (n = 13, not Gaussian distributed) (p<0.01), 48–72 hr APF 13.4 ± 1.07 SD (n = 10, non-Gaussian distribution). Non-parametric Kruskal–Wallis Test. (**E'**) Confirmation of the requirement of Notch signaling during pupation using tubGal80$^{ts}$: Control 11.71 ± 0.92 SD (n = 17, non-Gaussian distribution), 24–48 hr APF 15.55 ± 1.04 SD (n = 17, non-Gaussian distribution) (p<0.001), 36–60 hr APF 15.60 ± 1.17 SD (n = 18, Gaussian distribution) (p<0.001), 0–96 hr APF 14.15 ± 1.11 SD (n = 13, non-Gaussian distribution) (p<0.01). (**F–I**) comparison of axonal pattern formation at two different time points under control and gain of Notch activity. Scale bar: 20 µm. At 32 hr APF the axonal extension process is still active. Under both conditions (control (**F**) and gain of Notch function (**H**)) axons extend toward the medulla. (**G, I**) Axonal pattern of the DCNs at adult stage. While the control brain shows the normal pattern of medulla axons (**G**), under gain of Notch function there is only one medulla axon visible (**I**). (**J**) Quantification of the number of medulla axons using gain of Notch function and gain of Su(H) function. Activation of the canonical Notch pathway decreases the number of medulla axons significantly in comparison to the control. Control 12.33 ± 0.89 SD (n = 12, Gaussian distribution), gain of Notch function 2.33 ± 1.24 SD (n = 15, non-Gaussian distribution) (p<0.001), gain of Su(H) function 5.22 ± 1.99 SD (n = 9, Gaussian distribution) (p<0.05) (**E, E', J**: non-parametric Kruskal Wallis Test, Data shown: mean ± SEM). APF: after puparium formation; DCN: dorsal cluster neurons; NICD: intracellular domain of Notch; SD: standard deviation; SEM: standard error of the mean.

classical process of mutual inhibition, which specifies early progenitor cell fate, we demonstrate that mutual inhibition is reused in postmitotic DCN neurons. Three arguments support this interpretation. First, the decision to target the medulla neuropil of single cell Notch null mutant clone is probabilistic (~60%) rather than absolute (100%) as would be expected for cell fate specification. Second, the requirement for Notch signaling in establishing the wiring diagram is restricted to 24–60 hr APF, and third, Notch signaling affects neither the lineage size nor the expression of different cell fate markers in the DCNs early in development. It should be noted that medulla vs lobula axon target choice can be viewed as a form of sub-fate choice. The important discovery here is that this 'sub-fate' is not a cell-autonomous translation of birth order instructions, but is rather subject to regulation by the choices of neighboring cells. In that sense, it extends the timing and plasticity of the phase during which neurons can be said to acquire their 'fate'.

Here, we report lateral inhibition as a novel mechanism for brain wiring. It is likely that the inhibitory interactions are taking place between the neuronal cell bodies, especially given the requirement for the nuclear factors Su(H) and Mam and the effects on the E(spl) transcriptional reporter. However, interactions between axons prior to or shortly after their defasciculation within the medulla cannot be formally ruled out. Axonal target choice resulting from mutual inhibition by neighboring postmitotic neurons is interesting, not only as a novel function for Notch signaling but also as an additional layer of regulation that complements, but is independent of, traditional axon guidance cues. While it may be sensible for sensory neurons to couple fate to targeting so that the same sensory information is carried by the same type of neuron in different animals, it is reasonable to speculate that a degree of circuit-level plasticity is a more useful strategy in higher order brain centers, where meaningful information is likely to be encoded at the network level and not by single cells. It would be interesting to test if other patterns of alternative target choices by similar neurons, such as retinal projections into the midbrain (*Brown et al., 2000*), mammalian ocular dominance columns (*Crowley and Katz, 2002*), or the wiring variability of *Drosophila* olfactory interneurons (*Chou et al., 2010*) utilize similar mechanisms. Finally, given the inevitable and significant genetic and environmental variability in the real world, it seems rather intuitive that a mechanism that ensures the reproducibility of the connectivity pattern of an entire circuit, would be evolutionarily favored over a mechanisms that would risk the population connectivity pattern in favor of attempting to generate the same connectivity of precisely the same cell.

## Materials and methods

### Fly genetics

Flies were raised at 25°C or 28°C, except for temporal Notch activity experiments for which flies were raised at 18°C and shifted to 28°C during pupation for a specific time window of either 24 hr or until eclosion. For MARCM, crosses were set up at 25°C. 2 to 4 days after egg laying, samples were heat-shocked for 3 hr at 37°C and shifted back to 25°C until eclosion. Axonal pattern analysis was done on freshly eclosed adult flies.

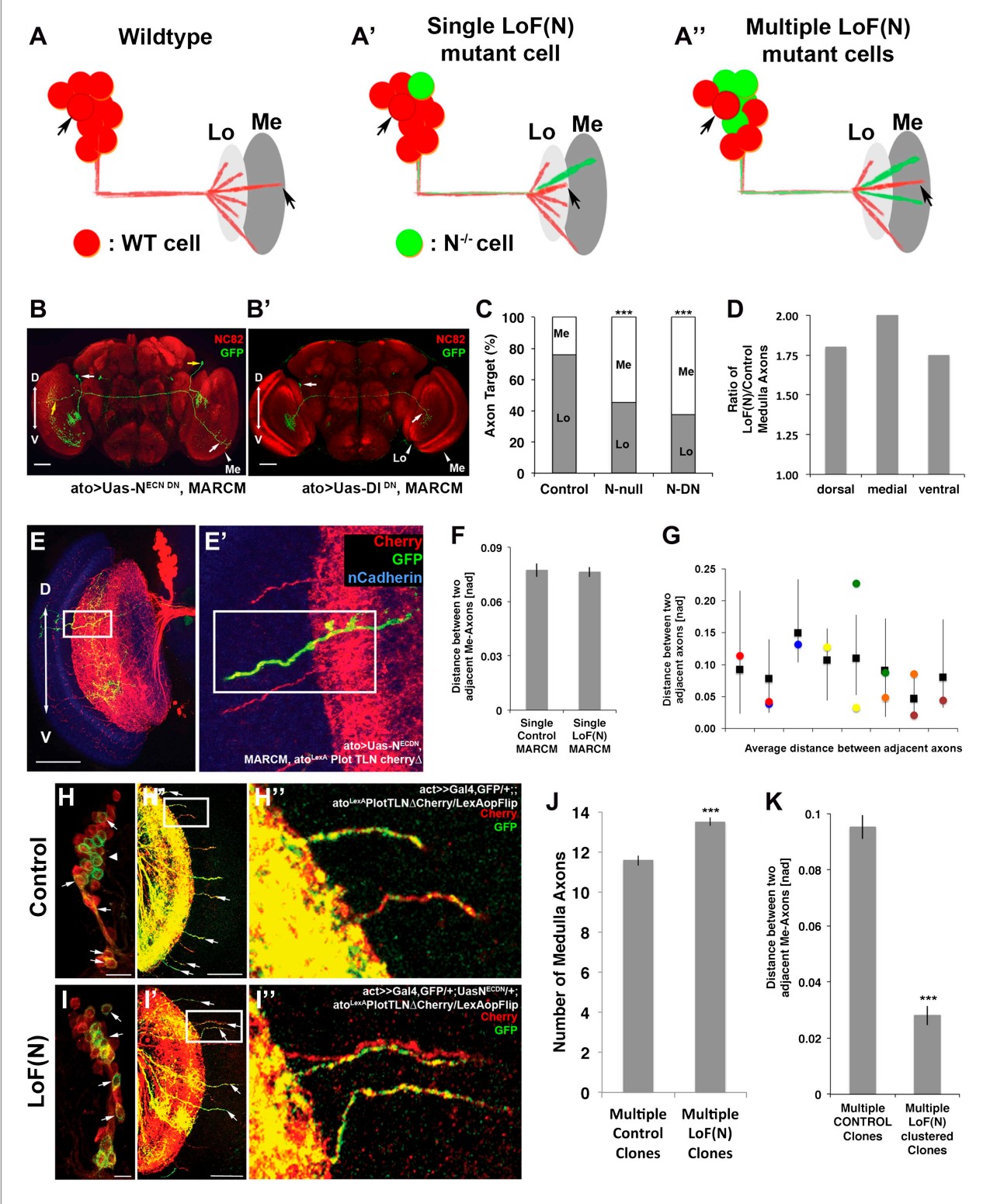

**Figure 8**. Notch cell autonomously regulates axon targeting. (**A–A''**) Schematic of a mutual inhibition hypothesis of DCN axon sorting under control and Notch loss of function conditions. (**A**) Under control conditions only single axons innervate the medulla neuropil from each interacting subcluster. The red cell represents one of the wild-type neurons. (**A'**) A single Notch mutant cell (green cell) will project its axon towards the medulla neuropil and consequently adjacent cells will retract back towards the lobula to ensure single medulla innervations. (**A''**) In contrast to the single Notch mutant cell situation, multiple Notch mutant cells will not compensate and generate multiple clustered medulla axons. (**B**) Adult *Drosophila* fly brain in which the neuropil is

*Figure 8. Continued on next page*

*Figure 8. Continued*

marked with mAb nc82 (red). Single GFP-positive Notch mutant DCNs is generated using the MARCM technique. White and yellow arrows mark DCN soma and their respective medulla axons. Scale bar: 50 μm. (**B'**) Single GFP-positive Delta mutant DCN is generated using the MARCM technique. White arrows mark DCN soma and their respective lobula axon. Scale bar: 50 μm. (**C**) Analysis of the MARCM experiment represented in (**B**). Approximately 24% of all control single cell clones project toward the medulla (n = 33). In contrast, Notch mutant single cell clones using either a null allele (N$^{55e11}$) or Notch$^{DN}$ innervate the medulla 55% or 63% of the time (n = 22 and 24), respectively (Control vs N$^{55e11}$, p<0.001; Control vs Notch$^{DN}$, p<0.001; Fisher's exact test). (**D**) The probability of a single DCN with no Notch activity to target the medulla is twice as high medially in comparison to the control, 1.8 times higher dorsally and 1.75 times higher ventrally (n = 22). (**E–E'**) DCN cluster labeled with the general membrane marker TLN$^{A}$Cherry (***Nicolai et al., 2010***) (red) driven by the LexA operon (LexAop) and a LexA knock-in into the *atonal* locus using the IMAGO (***Choi et al., 2009***) technique. A single GFP-positive (green) cell clone expressing Notch$^{DN}$ was generated using MARCM technique in this background. The mutant GFP-positive medulla axon does not show clustering with neighboring control axons (red). Scale bar: 50 μm. (**F**) Comparison of distances between two adjacent medulla axons: Single control MARCM 0.077 ± 0.048 SD (n = 189, not Gaussian distributed), single loss of Notch function MARCM 0.076 ± 0.051 SD (n = 177, non-Gaussian distribution) (p>0.05, Mann–Whitney Test). (**G**) Individual distances between six loss of Notch function single MARCM clones and their two adjacent medulla axons in comparison to median control distances. The distances of 5/6 loss of Notch function medulla axons (colored squares) are within control range (black squares + range bars). Black squares represent the average distance between two adjacent medulla axons, whereas the bar represent minimum and maximum distance between two neighboring medulla axons under control condition. (**H–H'', I–I'**) Multiple in vivo DCN soma were mutated for Notch activity using a combination of IMAGO knock-in of LexA into the *atonal* locus and the flip-out technique to activate Gal4 stochastically during DCN axon outgrowth. An example of multicell DCN clones under control (**H–H''**) and loss of Notch function conditions (**I–I'**). While the control clone displays single medulla axons (**H', H''**), clustered medulla axons are visible under loss of Notch function condition (**I', I''**). The D-V axis correlation between a DCN soma and its axon is maintained under loss of Notch function conditions (white arrows). (**J**) A comparison between multiple control and multiple loss of Notch function clones demonstrates that DCN patterns with clustered axons show a significant increased number of medulla axons. Multicell control clones 11.6 ± 1.19 SD (n = 20, non-Gaussian distribution), multicell loss of Notch function clones 13.65 ± 1.09 SD (n = 20, non-Gaussian distribution) (p<0.001, Mann–Whitney Test). Data shown: mean ± SEM. (**K**) Comparison of distances between two adjacent axons considering only GFP-positive axons under multiple control and multiple loss of Notch function clones. Multiple control clones: 0.09 [nad] (=normalized arbitrary distance) ± 0.053 (n = 165, non-Gaussian distribution), multiple loss of Notch function clones: 0.028 ± [nad] 0.015 (n = 19, Gaussian distribution) (p<0.001, Mann–Whitney Test). DCN: dorsal cluster neurons; D-V: dorsal–ventral; SD: standard deviation; SEM: standard error of the mean.

The following figure supplements are available for figure 8:

**Figure supplement 1**. Analysis of medulla axons in a genetic MARCM background.

**Figure supplement 2**. Loss of Notch function does not alter the D-V position.

**Figure supplement 3**. Multi-cell loss of N clone provides clustered axons.

The following flystrains were used:

*Figure 1*: (A–A''') *w/+;UAS-mCD8::GFP/+;atoGal4-14a,UAS-LacZ/+;*
*Figure 2*: (A–B') *w/+;UAS-mCD8::GFP/+;atoGal4-14a,UAS-LacZ/+;* (C, D) *w/+;;atoGal4-14a,UAS-CD8::GFP/UASLacZ.NZ;* (E–H) *yw, hsFlp/w;Sp/+;UAS FRT CD2, y FRT mCD8::GFP/+;atoGal4-14a,UASLacZ/+;*
*Figure 3*: *w/+;UAS-mCD8::GFP/+;atoGal4-14a,UAS-LacZ/UAS-RedStinger;*
*Figure 4*: (A) *w/+;UAS-mCD8::GFP/+;atoGal4-14a,UAS-LacZ/+;* (B) *w ;+;atoGal4-14a,UAS-mCD8::GFP/neurlacZ$^{A101}$;;* (C–F') *w;UAS-m-CD8::GFP/E(spl)mβ-CD2;atoGal4-14a,UAS-LacZ/+;*
*Figure 5*: (D, K, L) *w/+;UAS-mCD8::GFP/+;atoGal4-14a,UAS-LacZ/+;* (E, K, L) *w/+;UAS-mCD8::GFP/UAS-Notch$^{EC}$;atoGal4-14a,UAS-LacZ/+;* (F, L) *w/+;UAS-mCD8::GFP/UAS-D$^{de}$;atoGal4-14a,UAS-LacZ/+;* (G, L) *w/+;UAS-mCD8::GFP/Neur RNAi; atoGal4-14a,UAS-LacZ/+;* (H, M) *w/+;UAS-mCD8::GFP/Su(H) RNAi;atoGal4-14a,UAS-LacZ/+;* (I, M) *w/+;UAS-mCD8::GFP/Mam RNAi;atoGal4-14a,UAS-LacZ/+;* (J, M) *w/+;Mam RNAi /Su(H) RNAi;atoGal4-14a,UAS-LacZ/+;* (L) *w/ N RNAi;UAS-mCD8::GFP/+;atoGal4-14a,UAS-LacZ/+; w;UAS-mCD8::GFP/+;atoGal4-14a,UAS-LacZ/UAS-Ser$^{DN}$; w;UAS-mCD8::GFP/ UAS-D$^{de}$;atoGal4-14a,UAS-LacZ/UAS-Ser$^{DN}$;*
*Figure 6*: (A, A', D) *w/+;UAS-mCD8::GFP/+;atoGal4-14a,UAS-LacZ/+;* (B, B', D) ) *w/+;UAS-mCD8::GFP/UAS-Notch$^{EC}$;atoGal4-14a,UAS-LacZ/+;* (C, C', D) *w/+;UAS-mCD8::GFP/UAS-Pak$^{myr}$;atoGal4-14a,UAS-LacZ/+;* (D) *w/+;UAS-mCD8::GFP/Pak RNAi;atoGal4-14a,UAS-LacZ/+; bsk DN/+;UAS-mCD8::GFP/+;atoGal4-14a,UAS-LacZ/+; bsk DN/+;UAS-mCD8::GFP/ UAS-Notch$^{EC}$;atoGal4-14a,UAS-LacZ/+; w/+; Pak RNAi/ UAS-Notch$^{EC}$;atoGal4-14a,UAS-LacZ/+; bsk*

*DN/+;UAS-mCD8::GFP/ UAS-Pak$^{myr}$;atoGal4-14a,UAS-LacZ/+; (F) w/+;UAS-mCD8::GFP/UAS-Notch$^{Intra}$;atoGal4-14a,puc-LacZ/+; w/+;UAS-mCD8::GFP/+;atoGal4-14a,UAS-LacZ/+; w/+;UAS-mCD8::GFP/UAS-Notch$^{Intra}$;atoGal4-14a,UAS-LacZ/+;*

*Figure 7*: (C–E) N$^{ts}$/w; UAS-CD8::GFP/+;atoGal4-14a,UAS-LacZ/+; (E') w/N RNAi;tubGal80ts/+;atoGal4-14a,UAS-mCD8::GFP/+; (F, G, J) w/+;UAS-mCD8::GFP/+;atoGal4-14a,UAS-LacZ/+; (H, I, J) w;UAS-mCD8::GFP/UAS-N$^{Intra}$;atoGal4-14a,UAS-LacZ/+; (J) w/+;UAS-mCD8::GFP/UAS Su(H);atoGal4-14a,UASLacZ/+;

*Figure 8*: (B–D) FRT19A/FRT19A,hsFlp,tub-Gal80;UAS- GFP,UAS-nLacZ/UAS-Notch$^{EC}$;atoGal4-14a,UAS-mCD8::GFP/+; (B') FRT19A/FRT19A,hsFlp,tub-Gal80;UAS- GFP,UAS-nLacZ/ UAS-D$^{de}$;atoGal4-14a,UAS-mCD8::GFP/+; (C) FRT19A/FRT19A, hsFlp, Gal80;UAS- GFP,UAS-nLacZ/+;atoGal4-14a,UAS-mCD8::GFP/+; (C, D) FRT19A,N$^{55e11}$/FRT 19A, hsFlp,tub-Gal80;UAS- GFP,UAS-nLacZ/+;atoGal4-14a,UAS-mCD8::GFP/+; (E–G) FRT19A/FRT19A, hsFlp,tub-Gal80;UAS- GFP,UAS-nLacZ/UAS-N$^{EC}$;atoGal4-14a,UAS-mCD8::GFP/ato$^{LexA}$,LexAop-TLNΔCherry; (H–H", J, K) act5C >y >Gal4,UAS-mCD8::GFP/+;;ato$^{LexA}$,LexAop-TLNΔCherry/ LexAopFlp(N3); (I–I", J, K) act5C>y>Gal4,UAS-mCD8::GFP/+;UAS-Notch$^{EC}$/+;ato$^{LexA}$,LexAop-TLNΔCherry/LexAopFlp(N3);

*Figure 2—figure supplement 1*: (top) w/+;UAS-mCD8::GFP/+;atoGal4-14a,UAS-LacZ/UAS-RedStinger; (bottom) yw, hsFlp/w;Sp/+;UAS FRT CD2, y FRT mCD8::GFP/+;atoGal4-14a,UASLacZ/+;

*Figure 3—figure supplement 1*: w/+; +;atoGal4-14a,UAS-GFP,UAS-RedStinger/+;

*Figure 3—figure supplement 2*: (top) w/+; +;atoGal4-14a,UAS-GFP,UAS-RedStinger/+; (bottom) w/+;UAS-mCD8::GFP/+;atoGal4-14a,UAS-LacZ/UAS-RedStinger;

*Figure 5—figure supplement 3*: Dicer2/w; UAS-mCD8::GFP/+; atoGal4-14a,UAS-LacZ/+; Dicer2/w; UAS-mCD8::GFP/Delta ds RNAi; atoGal4-14a,UAS-LacZ/+;

*Figure 5—figure supplement 4*: w/+;UAS-mCD8::GFP/+;atoGal4-14a,UAS-LacZ/UAS-RedStinger; w/+;UAS-mCD8::GFP/UAS-Notch$^{EC}$;atoGal4-14a,UAS-LacZ/UAS-RedStinger;

*Figure 5—figure supplement 5*: w/+;UAS-mCD8::GFP/UAS-Notch$^{EC}$;atoGal4-14a,UAS-LacZ/+;

*Figure 8—figure supplement 1*: FRT19A/FRT19A, hsFlp,tub-Gal80;UAS- GFP,UAS-nLacZ/+;atoGal4-14a,UAS-mCD8::GFP/ato$^{LexA}$,LexAop-TLNΔCherry; FRT19A/FRT19A, hsFlp,tub-Gal80;UAS- GFP,UAS-nLacZ/UAS-N$^{EC}$;atoGal4-14a,UAS-mCD8::GFP/ato$^{LexA}$,LexAop-TLNΔCherry;

*Figure 8—figure supplement 2*: FRT19A/FRT19A, hsFlp,tub-Gal80;UAS- GFP,UAS-nLacZ/ UAS-N$^{EC}$;atoGal4-14a,UAS-mCD8::GFP/ato$^{LexA}$,LexAop-TLNΔCherry;

*Figure 8—figure supplement 3*: act5C>y>Gal4,UAS-mCD8::GFP/+;UAS-Notch$^{EC}$/+;ato$^{LexA}$,LexAop-TLNΔCherry/LexAopFlp(N3);

## Immunohistochemistry

Dissected larval, pupal, or adult brains were stained with specific primary antibodies using the protocol described (*Hassan et al., 2000*) and incubated with Alexa fluorescent secondary antibodies. Pictures from *Drosophila* brains were taken using confocal microscopy Nikon AIR confocal unit mounted on a TI2000 inverted microscope (Nikon Corp., Tokyo, Japan).

## Primary antibodies

The following antibodies and dilutions were used: mouse anti-ACJ6 (1:10), rabbit anti-Ato (1:2000), mouse anti-Delta (1:20), rabbit anti-Notch (1:1000), rabbit anti-Numb (1:200), rabbit anti-DsRed (1:500), rabbit anti-GFP (1:1000), mouse anti-GFP (1:1000), mouse anti-LacZ (1:1000), mouse anti-NC82 (1:100), rat anti-nCadherin (1:20), and mouse anti-CD2 (1:5000).

## Microscopy and image analysis

*Drosophila* brains were imaged using confocal microscopy Nikon AIR confocal unit mounted on a TI2000 inverted microscope (Nikon Corp.). The Nikon microscope was equipped with a Plan Apo 20× (0.72 NA) and a Plan Apo60× oil immersion (NA 1.40) objective lens. ImageJ was used to create Z-projections from confocal stacks. Images such as single or merged figures were created using Photoshop. For ease of viewing, images were rotated so that the DCN cluster is located on the top left corner of every image. To analyze the expression pattern of the E(spl)-CD2 carefully within the DCNs, DCN soma were isolated from the background using ImageJ. Confocal sections revealing individual DCN soma were obtained and CD2 fluorescence intensity within the different DCNs was analyzed.

## Statistical analysis

Data were presented as mean ± SEM. Results were analyzed using either Bonferroni or Kruskal–Wallis depending whether samples were normal distributed (Komogorov–Smirnoff test). In case of a single comparison, the Student's t-test or the Mann–Whitney test was used depending on the result of the normality test. Fisher's exact test was used for cases of contingency tables. For all statistical tests, GraphPad Prism was used.

## Positional data of neurons and axons

To generate profiles of DCN soma and their medulla axons, each object of interest, neuron or axon, was tagged with a 3D coordinate (proximal–distal, dorsal–ventral, anterior–posterior axis) scanning through a stack of confocal sections using ImageJ-*Cell Counter* module (*Figure 3— figure supplement 1*). The 3D coordinates were eventually normalized according to cluster diameter or lobula length and saved as *DCN profiles*. While analyzing the confocal sections, the number of neurons and medulla innervations were counted as they are a result of the number of coordinates. To calculate neuron to medulla axon ratio, the medulla axon profile was divided into 10 regions with the 'clusterdata' function in MATLAB. The size of each of the 10 units depends on the axonal pattern as determined by the cluster algorithm. The same division is then projected onto the soma cluster that generated the particular axonal pattern, to obtain 10 identical DCN soma divisions.

## Computational model

The computational model is implemented according to *Collier et al. (1996)* and is described below in detail.

## Equations

$$\frac{dN}{dt} = p_n \frac{dl^k}{a + dl^k} - \mu N,$$

$$\frac{dDl}{dt} = p_d \frac{1}{1 + bN^h} - \rho Dl,$$

$$dl = \frac{1}{r} \sum_i Dl.$$

The first function describes Notch activity of a cell over time. Notch activity of each cell is positively influenced by Delta activity in the adjacent cells. The second equation defines the intracellular level of Delta, which is influenced by Notch activity within the same cell. The third function computes Delta activity of adjacent cells. N and Delta represent Notch and Delta activity in each cell, whereas *dl* refers to the average Delta activity calculated based on adjacent cells. Notch and Delta activity are computed by functions parameterized by *a*, *b*, *k*, and *h*. The production rates are symbolized by the variables $p_n$ and $p_d$, whereas $\mu$ and $\rho$ represent decay rates.

## Settings of variables

In our simulation, we set the parameters to compute the differential equations described above as follows: $a = 0.01$, $b = 100$, $k = 2$, $h = 2$, $\mu = 1$, $\rho = 1$, and $p_d = 1$. $p_n$ depended on the signal strength of Notch signaling.

## Initial settings

Initially Notch and Delta activity levels were set to its signal strength, which were either 100% to simulate control condition or 10% to resemble downregulation of the transgene of interest. We assumed that 10% in case of a reduced Notch activity level will reflect the in vivo condition best (in vivo 18.4 ± 2.22 SD vs in silico 17.29 ± 3.96 SD) since Notch[DN] transgene for example is a strong repressor of Notch activity. We also tested in silico downregulation of Notch to 30%, and this results in approximately 20.33 ± 6.91 SD medulla axons. Given that Notch[DN] is a strong repressor of Notch function, 10% was chosen to simulate an in vivo loss of Notch function. Additionally, a noise term of 0–1% was introduced to the initial settings of Notch and Delta signaling.

### Detection of adjacent DCN soma

To determine the trans-Delta expression level of each DCN soma, all adjacent cells needed to be identified. Since *Collier et al. (1996)* used cells arranged in a hexagonal lattice, each cell had at most six adjacent cells, thus creating a subcluster of maximal seven cells. The DCN cluster consists of ~40 DCN soma, which are arranged in three dimensions. To divide the DCN cluster in subclusters of an average size of seven cells, Euclidean 3D distances between DCN soma were examined. DCNs were split into subclusters of about seven cells when we applied an Euclidean 3D distance of 0.17 [nad] (=normalized arbritrary distance) (*Figure 5—figure supplement 2*). Additionally, this also resulted in approximately 12 medulla axons (*Figure 5—figure supplement 3*).

### Computational simulation

To solve the differential equations described above, the Runge–Kutta (2, 3) formula was used for a timespan of 1000 time steps. During each timestep, each DCN soma was updated in its Notch and Delta expression level in three steps. First, 'trans-Delta' expression was determined. For this, Delta expression of adjacent DCN soma was detected using 0.17 [nad] as an Euclidean distance.

Second, Notch activity level of a given DCN was updated based on the average Delta expression in adjacent cells. Finally, Delta expression of a given DCN is recalculated based on Notch activity in that cell. These calculations were performed for each DCN soma and repeated a 1000 times.

After solving the different equations, the final Notch expression level of each cell was analyzed. For each subcluster, the DCN soma with the lowest Notch activity was determined since these DCN soma will project its axon toward the medulla. Under wild-type conditions, each subcluster has on average only one DCN soma with the lowest Notch activity, whereas under loss of Notch signaling, multiple DCN soma are predicted to have equally low Notch activity levels. Matlab Version 7.6.0.324 (R2008a) was used for computational simulations and predictions.

### Application of the mathematical model

The mathematical model is applied to analyze different aspects of the DCNs. First, the mathematical model should predict under control and loss of Notch activity condition the number of medulla axons to a certain degree. For this purpose, 16 different in vivo DCN clusters with their normalized 3D coordinates were taken as input for the mathematical model. These 16 DCN clusters were simulated a 1000 times. From these resulting 1600 data points, the average number of medulla axons, the standard deviation, and the standard error of the mean were calculated. Second, to analyze the final Notch activity pattern within an in vivo DCN cluster simulating either the control or loss of Notch condition, a single DCN cluster was applied to the mathematical model and presented in a 3D chart using different colors to symbolize lobula vs medulla DCN neurons (*Figure 5A,B*).

## Acknowledgements

We thank Gerald M Rubin for providing the unpublished LexAop-Flp lines, Utpal Banerjee for E(spl)-CD2 flies, Francois Schweisguth for Delta and *neur* RNAi flies, Jürgen Knoblich for the Numb antibodies, the Bloomington Stock Center and VDRC for *Drosophila* Stocks. We thank KU Leuven for providing the Software (Matlab) and Hardware that enabled computational analyses. We are very grateful to Sebastian Munck and the LiMoNe imaging facility for help with confocal microscopy. Confocal imaging was made possible through a Hercules Type 1 AKUL/09/037 grant coordinated by Wim Annaert. We thank Patrik Verstreken, Carmen Ruiz de Almodovar, and Ana Martin-Villalba for comments on the manuscript as well as Pierre Vanderhaeghen and members of the Hassan lab for stimulating discussions.

## Additional information

### Funding

| Funder | Grant reference number | Author |
| --- | --- | --- |
| Vlaams Institut voor Biotechnologie | | Bassem Hassan, Dietmar Schmucker |
| University of Leuven | | Bassem Hassan, Dietmar Schmucker, Yves Moreau |

| Funder | Grant reference number | Author |
|--------|------------------------|--------|
| Fonds Wetenschaplijk Onderzook | G.0543.08, G.0680.10, G.0681.10, G.0503.12, 1.2888.11N | Bassem Hassan, Marta Koch, Dietmar Schmucker, Yves Moreau |
| Belgian Science Policy | P6/20, P5/22 | Bassem Hassan, Dietmar Schmucker |
| agentschap voor Innovatie door Wetenschap en Technologie | 060729 | Marta Koch |
| Howard Hughes Medical Institute | | Bassem Hassan, Barret D Pfeiffer |

The funders had no role in study design, data collection and interpretation, or the decision to submit the work for publication.

## Author contributions

ML, Conception and design, Acquisition of data, Analysis and interpretation of data, Drafting or revising the article; MK, Acquisition of data, Analysis and interpretation of data; JY, NDG, M-LE, Acquisition of data; BDP, Contributed unpublished essential data or reagents; DS, Analysis and interpretation of data, Drafting or revising the article; YM, Analysis and interpretation of data; BAH, Conception and design, Analysis and interpretation of data, Drafting or revising the article

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
