## [Decision Letter]

Thank you for choosing to send your work entitled “Mutual inhibition among postmitotic neurons regulates robustness of brain wiring” for consideration at *eLife*. Your article has been evaluated by a Senior editor and 3 reviewers, one of whom is a member of our Board of Reviewing Editors.

The Reviewing editor and the other reviewers discussed their comments before we reached this decision, and the Reviewing editor has assembled the following comments based on the reviewers' reports.

Langen et al describe a developmental mechanism of neural wiring in the *Drosophila* visual system in which antagonistic interactions, via Notch signaling, in post-mitotic neurons, define distinct axon target choices within a lineage of apparently identical neurons. This is an interesting finding, which is globally well supported by the data, and which reveals a novel non-deterministic strategy by which reproducible nervous system connectivity is achieved.

The paper investigates how a single cluster of ∼38 neurons generated from a single neuroblast, and apparently molecularly identical, can still manage to have dramatically different projection patterns. The Dorsal Cluster Neurons (DCN) can either project to the medulla or to the lobula.

Very careful quantification of the projections and cell bodies indicates that there is extensive variability in the number of DCN produced from the unique neuroblast, as well as variability in the number of clusters along the dorso-ventral axis, but surprisingly a light level of stereotypy in the total number of neurons that project to the medulla.

The investigators explain how this is possible through a re-utilization of the same pathway used in lateral inhibition during the formation of embryonic neuroblasts, also in *Drosophila*. This is surprising as this Notch pathway acts as a lateral inhibitor only in the early steps of neurogenesis during embryogenesis, while we are dealing here with post-embryonic neurogenesis. One would have therefore expected a mode of lateral interactions mediated by asymmetric division of cells and asymmetric distribution of Numb, as in bristle precursor cells. The authors therefore show that, from each of the clusters of DCNs, only a fixed number of neurons, inversely proportional to the size of the cluster, choose to target the medulla rather than the lobula. Therefore, we are dealing with an embryonic type of Notch function rather then a post-embryonic mode.

Points to address:

* The authors argue repeatedly that the Notch choice is independent of the fate of the two populations of cells. However, this is far from being demonstrated as only two molecular markers are shown to be identical (Acj6 and Ato). It would take a lot more work to demonstrate that these are one type of neuron with different projection patterns. In fact, it is highly likely that the two cell populations are different from one another since they project differently, target different neurons, and might have different neurotransmitters. How can the authors be sure that there are not in fact two subsets of DCN neurons and that Notch functions classically in fate determination? This should be discussed.

* The authors address when Notch is required, but it is unclear where Notch functions and this could be clarified either with new data or at least in the Discussion. Is the Delta/Notch signaling functioning in the axons themselves (is Notch detectable in this compartment), or within the soma as implied indirectly by the relationship established early in the manuscript between DCN soma clustering and axonal target choice?

* The modeling is nice and does not detract from the quality of the paper. However, it is not necessary to use equations to predict that loss of Notch will lead to an increase in the proportion of medulla projecting neurons. Anyone who knows Notch would have predicted this.

* One reason for nature choosing to use the lateral inhibition system of the embryonic neurogenesis rather than the N on/N off Numb-biased choice might simply be the goal to achieve: in one case, a few neuroepithelial cells in the embryo decide to become neuroblasts while the majority remains epithelial (a case similar to the DCNs where the majority of neurons retracts to the lamina). In contrast, the Numb-mediated Notch function leads to 50-50 neurons A vs neuron B. In both cases, the neurons are different and it is likely that it is the same thing for DCN. Because the choice is made late, the common “master control” transcription factors establishing the lineage remain on late. But it is likely that a profiling of medulla vs lobula DCN neurons would lead to significant profile differences.

* Figure 8 contains several issues: for example, Figure 8h, it would be more illustrative to show an example in which a similar number of axons are labelled with GFP as in Figure 8i, because they claim that in the experimental condition the mutant (green) axons innervate the medulla, unlike the case in the control; but, because the control they show labels all the axons in green, it makes it difficult to appreciate the result, and prove that the effect is not solely due to the genetic flip-out manipulation in the neurons, but rather specifically due to the lack of Notch.

Figure 8i shows a cluster of one mutant (green) and one wild-type (red) axon, which is counter to their model that wild type axons compensate for the presence of mutant axons by restricting to the lobula – what is the explanation for this? Figure 8k: the distances between adjacent axons should be calculated for all axons and not only for GFP positive axons.

---

## [Author Response]

** The authors argue repeatedly that the Notch choice is independent of the fate of the two populations of cells. However, this is far from being demonstrated as only two molecular markers are shown to be identical (Acj6 and Ato). It would take a lot more work to demonstrate that these are one type of neuron with different projection patterns. In fact, it is highly likely that the two cell populations are different from one another since they project differently, target different neurons, and might have different neurotransmitters. How can the authors be sure that there are not in fact two subsets of DCN neurons and that Notch functions classically in fate determination? This should be discussed*.

We agree, at least in part, with the reviewers. Even if the two DCN subpopulations were somehow intrinsically different, this difference is still subject to post-mitotic regulation by lateral inhibition, which is the important point here. Nonetheless to address the reviewers' valid concern, we have now added the following paragraph to the Discussion:

“It should be noted that medulla vs. lobula axon target choice can be viewed as a form of sub-fate choice. The important discovery here is that this “sub-fate” is not a cell-autonomous translation of birth order instructions, but is rather subject to regulation by the choices of neighboring cells. In that sense, it extends the timing and plasticity of the phase during which neurons can be said to acquire their “fate”.

** The authors address when Notch is required, but it is unclear where Notch functions and this could be clarified either with new data or at least in the Discussion. Is the Delta/Notch signaling functioning in the axons themselves (is Notch detectable in this compartment), or within the soma as implied indirectly by the relationship established early in the manuscript between DCN soma clustering and axonal target choice*?

We believe the requirement for nuclear factors makes it very likely that the competition is occurring between cell bodies. However, the reviewers are correct in that there is no formal way to genetically dissect the function in the axon from the function in the cell body. We have therefore added this text to the Discussion:

“It is likely that the inhibitory interactions are taking place between the neuronal cell bodies, especially given the requirement for the nuclear factors Su(H) and Mam and the effects on the E(spl) transcriptional reporter. However, interactions between axons prior to, or shortly after their defasiculation within the medulla cannot be formally ruled out.”

** The modeling is nice and does not detract from the quality of the paper. However, it is not necessary to use equations to predict that loss of Notch will lead to an increase in the proportion of medulla projecting neurons. Anyone who knows Notch would have predicted this*.

We do not agree with the reviewers on this point. First, it was far from intuitive that the existing model we used would have worked well in a 3D irregularly shaped cellular cluster. Second, this model had never been previously tested in Notch loss of function conditions. Third, and most importantly, the modeling does not merely predict an increase of medulla axon number, it does so quantitatively and with accurate spatial information. Thus, we believe the model is an important extension of the quantitative aspects of the biology reported in this paper.

** One reason for nature choosing to use the lateral inhibition system of the embryonic neurogenesis rather than the N on/N off Numb-biased choice might simply be the goal to achieve: in one case, a few neuroepithelial cells in the embryo decide to become neuroblasts while the majority remains epithelial (a case similar to the DCNs where the majority of neurons retracts to the lamina). In contrast, the Numb-mediated Notch function leads to 50-50 neurons A vs neuron B. In both cases, the neurons are different and it is likely that it is the same thing for DCN. Because the choice is made late, the common “master control” transcription factors establishing the lineage remain on late. But it is likely that a profiling of medulla vs lobula DCN neurons would lead to significant profile differences*.

A minor objection to the phrase “nature choosing” not withstanding, we essentially agree with the reviewers and this is the entire point of our paper. Lateral inhibition in this context is acting as a mechanism of developmental plasticity, allowing for at least one aspect of terminal fate choice to be open to competition and selection. In terms of the profiling likely to show intrinsic differences between medulla and lobula targeting neurons, this is highly likely to be the case. However, what is important is to ask whether such differences are consequent to Notch activity during axon target sorting. If they are, that would suggest that such molecular differences emerge only after that competition between cells is initiated. This would indeed be a very exciting project to pursue in the future when molecular profiling can be done on small cell populations in the fly brain. At any rate, we have now added the following paragraph to the Discussion:

“It should be noted that medulla vs. lobula axon target choice can be viewed as a form of sub-fate choice. The important discovery here is that this “sub-fate” is not a cell-autonomous translation of birth order instructions, but is rather subject to regulation by the choices of neighboring cells. In that sense, it extends the timing and plasticity of the phase during which neurons can be said to acquire their “fate”.”

** Figure 8 contains several issues: for example, Figure 8h, it would be more illustrative to show an example in which a similar number of axons are labelled with GFP as in Figure 8i, because they claim that in the experimental condition the mutant (green) axons innervate the medulla, unlike the case in the control; but, because the control they show labels all the axons in green, it makes it difficult to appreciate the result, and prove that the effect is not solely due to the genetic flip-out manipulation in the neurons, but rather specifically due to the lack of Notch*.

This is in fact the case in figure 8h and 8i. We count approximately 15 GFP positive cells in each panel.

*Figure 8i shows a cluster of one mutant (green) and one wild-type (red) axon, which is counter to their model that wild type axons compensate for the presence of mutant axons by restricting to the lobula – what is the explanation for this*?

We do not agree with this comment. When lateral inhibition breaks down, axon target choice defaults to the intrinsic levels of JNK activity, which now are no longer subject to differential regulation by Notch. Since JNK activity is deterministic to DCN axon target choice (Srahna et al., 2006), both wild type and Notch mutant axons may have similarly high levels of JNK activity and may target the medulla.

*Figure 8k: the distances between adjacent axons should be calculated for all axons and not only for GFP positive axons*.

Given that cells outside a mutated subcluster are not affected, it is not clear to us why axonal distances in these normal cells should be taken onto account. In both WT and Notch mutant cases we quantified only the distances within the GFP clones, which we believe is the relevant measurement to make.